# The Intersection of Health Rehabilitation Services with Quality of Life in Saudi Arabia: Current Status and Future Needs

**DOI:** 10.3390/healthcare11030389

**Published:** 2023-01-30

**Authors:** Abdullah M. Alanazi, Abrar M. Almutairi, Monira I. Aldhahi, Tareq F. Alotaibi, Hassan Y. AbuNurah, Lafi H. Olayan, Turki K. Aljuhani, Ahmad A. Alanazi, Marwh G. Aldriwesh, Hassan S. Alamri, Majid A. Alsayari, Abdulelah M. Aldhahir, Saeed M. Alghamdi, Jaber S. Alqahtani, Abdullah A. Alabdali

**Affiliations:** 1Department of Respiratory Therapy, College of Applied Medical Sciences, King Saud Bin Abdulaziz University for Health Sciences, Riyadh 11481, Saudi Arabia; 2King Abdullah International Medical Research Center, Riyadh 11481, Saudi Arabia; 3Research Unit, College of Applied Medical Sciences, King Saud Bin Abdulaziz University for Health Sciences, Riyadh 11481, Saudi Arabia; 4Department of Rehabilitation Sciences, College of Health and Rehabilitation Sciences, Princess Nourah Bint Abdulrahman University, Riyadh 11564, Saudi Arabia; 5Department of Anesthesia Technology, College of Applied Medical Sciences, King Saud Bin Abdulaziz University for Health Sciences, Riyadh 11481, Saudi Arabia; 6Department of Occupational Therapy, College of Applied Medical Sciences, King Saud Bin Abdulaziz University for Health Sciences, Riyadh 11481, Saudi Arabia; 7Department of Audiology and Speech Pathology, College of Applied Medical Sciences, King Saud Bin Abdulaziz University for Health Sciences, Riyadh 11481, Saudi Arabia; 8Department of Clinical Laboratory Sciences, College of Applied Medical Sciences, King Saud Bin Abdulaziz University for Health Sciences, Riyadh 11481, Saudi Arabia; 9Department of Rehabilitation Services and Programs, Sultan Bin Abdulaziz Humanitarian City, Riyadh 13571, Saudi Arabia; 10Respiratory Therapy Department, Faculty of Applied Medical Sciences, Jazan University, Jazan 45142, Saudi Arabia; 11Clinical Technology Department, Respiratory Care Program, Faculty of Applied Medical Sciences, Umm Al-Qura University, Makkah 21961, Saudi Arabia; 12Department of Respiratory Care, Prince Sultan Military College of Health Sciences, Dammam 34313, Saudi Arabia; 13Department of Emergency Medical Services, College of Applied Medical Sciences, King Saud Bin Abdulaziz University for Health Sciences, Riyadh 11481, Saudi Arabia

**Keywords:** rehabilitation, health services, quality of life, Saudi Arabia

## Abstract

Quality of life (QoL) is essential for maintaining a healthy, balanced lifestyle, especially among individuals with chronic diseases. Saudi Arabia (SA) launched a health sector transformation program as part of the nationwide Vision 2030 initiative to ensure the sustainable development of efficient healthcare services, aiming to improve health by increasing well-being and QoL. More investigation into the current status of health rehabilitation services provided to individuals with chronic diseases and future needs to optimize services and improve QoL is needed. This was narratively discussed by experts from different health rehabilitation services in SA. Comprehensive health rehabilitation services including orthopedic, occupational, cardiac, pulmonary, critical care, perioperative, hearing and speech, substance use disorders, and vocational rehabilitation services were addressed. Health rehabilitation services in SA, as in other countries, are suboptimal for individuals in health rehabilitation programs. To optimize the QoL of individuals with chronic diseases, health rehabilitation services should be tailored based on the unique requirements of each service and its serving patients. The shared need to improve health rehabilitation services includes the adoption of home-based and telehealth services, the integration of multi-governmental sectors, the empowerment and allocation of health rehabilitation specialists, public awareness campaigns, policy legislation and guideline development, and the implementation of a long-term follow-up system. This review is one of the first to address the intersection of health rehabilitation services and QoL in SA; urgent and holistic actions are paramount to address the pressing need to optimize SA’s health rehabilitation services. The experts’ recommendations in this study may be applicable to other countries’ health systems, as health rehabilitation services are not well optimized globally.

## 1. Background

Living a healthy, balanced lifestyle is essential for promoting a high quality of life (QoL) [1], particularly among individuals who are affected by chronic illnesses. The concept of QoL is a pivotal contributor to developing and supporting a healthy ecosystem in society [1]. In 2018, Saudi Arabia (SA) launched a health sector transformation program as part of the nationwide Vision 2030 initiative to ensure the sustainability and development of efficient healthcare services in the country [2]. This program is based on the principle of value-based healthcare, which ensures that transparency and financial sustainability are in line with improving patient outcomes by applying a new model of care related to disease prevention [2,3]. It ultimately aims to improve health by increasing the duration of well-being and QoL in SA [2].

Long-term conditions are the leading causes of morbidity and mortality worldwide. SA is not an exception, as the most important factors causing disability in the country are cancer, diabetes, cardiopulmonary diseases, injuries from road traffic accidents, and mental illness [4,5]. These health conditions can substantially affect individuals’ physical, psychosocial, and occupational functioning [6]. Most long-term conditions, especially those with multimorbidity, have no complete cure or recovery of physical, psychological, or social well-being [7], which may potentially compromise QoL. Therefore, long-term rehabilitation is at the core of patient care to enhance well-being and QoL [7,8].

The ability to perform ADLs can be considered when evaluating QoL [9]. Earlier research defined health-related quality of life (HR-QoL) as an individual’s personal perception of their physical, mental, environmental, and social health [10]. It is a state of well-being and consists of two components: the ability to perform ADLs related to physical, psychological, and social well-being, and individual satisfaction with the degree of operation and disease control [10]. The subjective evaluation of an individual’s HR-QoL is influenced by a multitude of factors ranging from an individual’s heritage, lifestyle, and life circumstances to family, work, and the broader social environment [10,11].

The prevalence and longevity of individual disabilities are increasing due to population growth, aging, and medical advancements that protect and increase longevity [12]. These factors contribute to the increasing demand for healthcare and rehabilitation services to minimize the progression of chronic health conditions before debilitating symptoms or life-threatening events occur [12,13]. The potential factors that influence QoL could be extrinsic, such as environmental and social factors, or intrinsic, such as individual factors (e.g., age, sex, body weight, mental health), lifestyle behavior (e.g., functional independence), health condition, and education [14]. Rehabilitation services that integrate diet, therapeutic exercise, behavioral modification, and other components have been shown to optimize health functioning, improve symptoms, enhance HR-QoL, and reduce the utilization of health care [15,16]. Recently, the Health Sector Transformation Program was established as part of the Kingdom’s Vision 2030 strategy. The program’s goals include expanding access to health care services by providing universal coverage, promoting geographic equity, and expanding the use of electronic medical records and other forms of e-health. A significant focus of this nationwide initiative is health innovation aimed at enhancing the quality and effectiveness of healthcare services, including rehabilitation [1]. However, the current status of each rehabilitation service, including any future needs to improve the services, and the QoL of people with chronic health conditions have not been fully established.

The purpose of this review is to highlight the current status of health rehabilitation services provided in SA and to identify the future needs to improve the services that intersect with QoL. This study was conducted by experts specializing in different health rehabilitation fields, including orthopedic, occupational, cardiac, pulmonary, critical care, perioperative, hearing and speech, substance use disorder, vocational rehabilitation, and healthcare services, to comprehensively assess the health rehabilitation services provided in SA. Ultimately, this review will assist policymakers, clinicians, and researchers in perceiving the current status and future needs of health rehabilitation services in the country to meet the nation’s 2030 vision for optimizing QoL. In addition, the experts’ recommendations in this study can be applied to the global context, as health rehabilitation services are not well optimized globally.

## 2. Orthopedic Rehabilitation

### 2.1. Current Status

Orthopedic rehabilitation is a professional practice that focuses on returning individuals to function and reversing their trajectory of disability. Rapid changes in orthopedic rehabilitation to accommodate the effective and efficient management of patients with musculoskeletal (MSK) conditions have been implemented. Orthopedic rehabilitation services can include a multidisciplinary team of physical and occupational therapists to provide efficient and relatively specialized rehabilitation. Orthopedic rehabilitation in SA is an efficient system that operates under the government of the Ministry of Health, which manages over 494 hospitals and 2300 primary healthcare centers [17]. According to the national statistics, patient satisfaction with hospital services increased from 79.9% in 2018 to 92.6% in 2020. Orthopedic rehabilitation services and rehabilitation sectors in general aligned their mission with the Saudi Vision 2030 to enhance HR-QoL.

In SA, musculoskeletal conditions are escalating rapidly, ranking in the top three potential reasons to visit the hospital and accounting for 38% of family practice visits [18]. The resultant effects of these conditions dictate individuals’ physical and mental health and impact social connectedness and emotional well-being. The musculoskeletal-related symptoms range from structural and functional impairment to activity limitation and participation restriction, all of which could influence the QoL. Furthermore, injuries and falls are considered the first and the third, respectively, in the rank of most the common causes of death and disability in 2019 [19]. Moreover, the Institute for Health Metrics and Evaluation showed that low physical activity is one of the risk factors contributing to disability-adjusted life years lost in SA [20].

Four dimensional factors that have an association with QoL include MSK pain, physical function, mental health, and the self-perception of health [18]. The quality of the rehabilitation services provided is a key measure of the patient’s QoL, and there is currently a growing interest in measuring HR-QoL. For instance, osteoarthritis is the most prevalent condition in SA and is one of the primary causes of pain and disability worldwide [21]. Physical therapy is the first line of intervention. Physical therapy interventions, including aquatic physical therapy [14,22] or overground training [23], showed a substantial improvement in patient QoL. Patient-reported outcomes, including the generic tools related to specific aspects of individuals’ health, are suggested to be used as a long-term follow-up for enhancing the recovery of the patients in rehabilitation. Therefore, the provision of QoL measures in research that focuses on physical therapy outcomes has been augmented. The overarching goal of physical therapy is to ensure pain-free movement, improve function, and implement psychosocial support to improve patients’ health and overall well-being.

### 2.2. Future Needs

Assessing QoL in orthopedic physical therapy is essential for evaluating patient well-being and intervention efficacy. The major limitation of the QoL measurement is a lack of sensitivity; therefore, a patient-centered QoL measurement can be established to address all the factors that impact recovery and perception of care. Furthermore, despite the widespread provision of orthopedic physical therapy for movement-related impairment, few studies have focused on the optimal parameters for exercise interventions designed to improve QoL. Therefore, it is essential to understand the current status of HR-QoL, which requires identifying the optimal exercise training parameters to optimize HR-QoL. Additionally, the predisposing risk factors for musculoskeletal disabilities require the imperative action of orthopedic physical therapy stockholders to enhance social awareness of physical activity and initiate health initiatives that will improve physical activity in the country. Furthermore, orthopedic physical therapy has become an area of considerable interest in most health rehabilitation centers in SA, particularly in secondary and tertiary health care. However, the availability of physical therapy services in primary healthcare centers must be evaluated to meet the growing health needs of the Saudi population, especially among individuals residing in rural areas.

## 3. Occupational Rehabilitation

### 3.1. Current Status

Occupational therapy is a professional practice in healthcare that applies the therapeutic use of individuals’ daily life occupations to improve occupational performance and participation [24]. The scope of occupational rehabilitation services include the use of clinical reasoning and judgment to evaluate and diagnose occupational issues and provide occupation-based interventions to reduce them. Habilitation, rehabilitation, and the promotion of physical and mental health wellness for individuals with all levels of occupational issues, such as chronic pain, sport and work injuries, and congenital and developmental disorders, are the main services that an occupational therapist provides. These services should be provided to clients who have or are at risk of developing an injury, morbidity, disability, or activity limitation [24].

Occupational rehabilitation services play a major role in improving the QoL of clients. Since the beginning of the occupational therapy profession, a link has been established between occupational rehabilitation and QoL. Occupational rehabilitation services have been shown to improve QoL by increasing ADLs, instrumental activities of daily living (IADLs), and community participation among individuals with occupational issues [25,26]. Multiple studies have revealed associations between greater ADLs, IADLs, and community participation restriction with greater QoL reduction. Occupational rehabilitation services have demonstrated improvements in QoL in various healthcare settings, which are routinely measured by different instruments such as the Barthel Index, Functional Independence Measure, The Canadian Occupational Performance Measure, and The Lawton IADLs scale [27,28,29,30,31,32]. In SA, the Ministry of Health is leading a national initiative in collaboration with the National Center for Developmental and Behavioral Disorders (NCDBD) which aims to introduce legislation and recommendations to improve detection and early intervention services to improve, rehabilitate, and optimize QoL for children with developmental and behavioral disorders [33].

### 3.2. Future Needs

Occupational rehabilitation services are the key to the NCDBD initiative [34]. However, occupational rehabilitation services still face obstacles. First, the occupational therapy profession does not have a clearly defined scope of practice, which may cause problems with understanding the importance of occupational rehabilitation in healthcare settings. Second, the number of occupational therapists is relatively low. This issue is consistent with other countries, such as the United States and Norway, as the demands for occupational therapists increases [35,36,37]. Indeed, it was estimated that occupational therapy had the national greatest need among healthcare professionals in SA, and 137 full-time equivalents are needed to meet the national need for occupational therapists in SA [38]. Thus, the productivity and implementation of occupational rehabilitation is affected [39]. Embedding occupational rehabilitation within healthcare facilities, which could be achieved through the recognition of occupational rehabilitation services with a defined scope of practice and the advanced training of occupational therapists, is highly needed to develop high-quality services and provide competent personnel to serve individuals with occupational disorders in SA.

## 4. Cardiac Rehabilitation

### 4.1. Current Status

Despite advanced and evolving interventions in medicine, cardiovascular disease is the global leading cause of mortality [40]. In SA, the estimated age of patients with acute and chronic heart failure was 10 years younger than patients in other developed countries, coupled with high-risk factors of cardiometabolic diseases such as coronary artery disease and diabetes mellitus [41]. Cardiac rehabilitation has emerged as an effective intervention that has been shown to reduce morbidity and mortality and improve QoL [42,43]. Cardiac rehabilitation is associated a with 42% lower risk of all-cause mortality [44]. Furthermore, cardiac rehabilitation has been shown to reduce hospital admissions, increase exercise capacity, and reduce associated disease symptoms [42,43]. However, cardiac rehabilitation is not limited to patient assessment, nutritional counseling, weight management, blood pressure management, psychological support, physical activity interventions, and smoking cessation [45]. There are recommended instruments that monitor patients’ improvements during and after cardiac rehabilitation programs, such as the Borg Rating of Perceived Exertion Scale, the multiple-repetition maximum test, the six–minute walk test, and the incremental shuttle walk test, which have been reflected in patients’ QoL [46].

Randomized controlled trials have shown strong evidence regarding the cost-effectiveness and benefits of cardiac rehabilitation. However, access to cardiac rehabilitation programs is poor [47]. In SA, the number of cardiac rehabilitation programs is low, and they are mostly run by cardiologists and physiotherapists to manage patients with heart failure and heart transplants. In addition, most of these programs are not specific to cardiac rehabilitation (general rehabilitation involving exercise training) [48]. Furthermore, these general programs face different obstacles and difficulties in conducting effective cardiac rehabilitation, including limited enrollment capacities and personnel, especially with the requirement of qualified specialists in cardiac rehabilitation [48].

### 4.2. Future Needs

There is a clear need to increase the number of cardiac rehabilitation programs in SA, as well as the number of personnel needed to run such programs. An awareness of the benefits of cardiac rehabilitation is warranted to motivate people with chronic heart disease to take up training and increase program retention. Furthermore, there is a need to increase individual referrals by cardiologists to cardiac rehabilitation programs and implement a long-term follow-up, which necessitates a comprehensive initiative by stakeholders to ease the enrollment and monitoring of program outcomes. All these factors are expected to increase lifespan and improve QoL among individuals with chronic cardiovascular disease.

## 5. Pulmonary Rehabilitation

### 5.1. Current Status

Pulmonary rehabilitation is an interdisciplinary program that is mostly run by pulmonologists, physiotherapists, and respiratory therapists for patients with chronic pulmonary diseases such as chronic obstructive pulmonary disease (COPD), interstitial lung diseases, bronchiectasis, asthma, and scoliosis. It is individually tailored and designed to improve an individual’s physical and social performance by delivering exercise and educational sessions to people with chronic respiratory diseases [49]. According to the Global Initiative for Chronic Obstructive Lung Disease, pulmonary rehabilitation is essential for managing the respiratory disease symptoms and physical disabilities caused by exercise intolerance [50]. There is strong evidence that pulmonary rehabilitation improves cardiovascular risk markers, such as blood pressure and arterial stiffness, in people with respiratory diseases [51]. In addition, pulmonary rehabilitation programs decrease dyspnea, increase exercise capacity, and improve HR-QoL. They also reduce the risk of hospitalizations through exercise training, psychosocial and nutritional interventions, education, and smoking cessation [52]. Patient outcomes that reflect QoL in pulmonary rehabilitation programs are routinely measured by a six-minute walk test, Chronic Respiratory Disease Questionnaire, St George’s Respiratory Questionnaire, and a 36-Item Short Form Survey [53,54,55].

Despite the benefits of pulmonary rehabilitation [50], the uptake and completion rates among people with chronic respiratory diseases are poor due to issues surrounding transportation, timing, and distraction from the patient’s usual routine [56,57]. A recent Cochrane review reported a 30% decline rate for people with COPD who were referred for pulmonary rehabilitation; 40% of those who attended the first session did not complete the program [56]. Although the prevalence of COPD in SA is lower than in other developed countries, the prevalence of COPD from 1990 to 2019 in SA increased by 49%, rising from 1381.26 cases per 100,000 people to 2053.04 cases per 100,000 people, while the incidence rate rose by 43.4%, increasing from 101.18 new cases per 100,000 in 1990 to 145.06 in 2019 [5]. The number of Saudis diagnosed with COPD in 2019 was estimated at 434,560.64, representing a growth of 329.82% from 1990. In 2019, COPD was responsible for 57% of all mortalities caused by chronic respiratory diseases and 1.65% (1.39–1.88) of the total number of deaths in SA [5,19]. This is expected to increase healthcare costs and the demand for tailored pulmonary rehabilitation for individuals diagnosed with COPD in the country. On the other hand, limited studies investigated the uptake of pulmonary rehabilitation in other chronic pulmonary diseases in SA. AlMoamary M.S. (2012), however, reported promising outcomes for a limited number of patients with interstitial lung diseases, bronchiectasis, asthma, and scoliosis after enrollment in a pulmonary rehabilitation program in SA [58].

Although the government has made great efforts to invest in the healthcare system, the number of pulmonary rehabilitation programs is limited [59]. A recently published study stated that there are a limited number of pulmonary rehabilitation programs for people living with COPD, highlighting that there is no pulmonary rehabilitation center in the Eastern Province of the country [59]. Due to these limitations, the burden of chronic respiratory diseases in the nation is expected to increase, necessitating immediate action.

### 5.2. Future Needs

There are various barriers to establishing efficient pulmonary rehabilitation programs in SA, including a lack of national guidelines to standardize the services, a lack of qualified and trained personnel, and hospital capacity [59]. Moreover, there is a clear need to establish a national organization under the Ministry of Health to monitor pulmonary rehabilitation and standardize its services. Furthermore, the awareness of the public and healthcare professional of the benefit of pulmonary rehabilitation is highly needed to increase the enrollment, retention, health-related outcomes for individuals affected by chronic respiratory diseases [60].

Tackling such obstacles to improving pulmonary rehabilitation uptake is imperative. One suggested strategy is telepulmonary rehabilitation using digital technology, which has been shown to have equal efficacy to center-based pulmonary rehabilitation [61]. This approach also has a high acceptance rate and low dropout rate, as demonstrated by Al ghamdi et al. [62]. The approach of telepulmonary rehabilitation in COPD management is increasing, providing considerable opportunities to help with symptom management, reduce health resource utilization, and improve health status [63]. Reducing healthcare cost utilization is another important consideration in which the telepulmonary rehabilitation approach could be used, potentially lowering costs while maintaining the quality of rehabilitation in community centers, decreasing the pressure on gyms and rehabilitation centers, delivering individualized care, and facilitating care coordination among healthcare professionals [63].

## 6. Critical Care Rehabilitation

### 6.1. Current Status

Generally, ICU critical illnesses are clinical syndromes and are heterogeneous by nature, thus making the reliability of critical illness epidemiological data unavailable and challenging to obtain. Acute respiratory distress syndrome, sepsis, and acute kidney injury are examples of critical illnesses that are treated in the ICU [64,65]. The vulnerability and complexity of critical care could inadvertently predispose patients to immobility and prolonged bed rest, leading to a condition of profound muscle weakness termed intensive care unit acquired weakness (ICU-AW) [66]. ICU-AW can occur as early as 24 h after ICU admission and affects more than one-third of ICU patients who receive mechanical ventilation for five days [67]. Higher mortality rates and delayed recovery outcomes are associated with ICU-AW [68]. Following discharge, ICU survivors experience physical, cognitive, and psychological impairments [67,69]. The six-minute walk distance at the two-year follow-up was significantly shorter in ICU survivors than in the normal population [67]. Depression, anxiety, post-traumatic stress disorder (PTSD), delirium, and cognitive dysfunction are also increasingly being recognized among ICU survivors [70]. These long-term, debilitating symptoms of ICU survivors can also extend to their family members and are collectively described as post-intensive care syndrome (PICS), which substantially affects their functional status and HR-QoL [67,69,71,72].

Nevertheless, evidence-based practice has shown that the impact of long-term ICU impairments can be mitigated by interdisciplinary rehabilitation during the ICU stay and after discharge [73]. Implementation of the ABCDEF bundle is an evidence-based guide that includes: assess, prevent, and manage pain (A); both spontaneous awakening trials (SAT) and spontaneous breathing trials (SBT) (B); choice of analgesia and sedation (C); delirium: assess, prevent, and manage (D); early mobility and exercise (E); and family engagement and empowerment (F) [74]. Implementing the ABCDEF bundle has been shown to be a potent approach to improving mortality, recovery of functional status, and QoL [75]. Of note, there are different instruments that are adapted to measure QoL among ICU survivors, such as the 36-Item Short Form Survey and the EuroQOL-5D quality of life assessment tool [55,76].

Recognizing the status of ICU rehabilitation is important for improving the services provided to ICU survivors in each country [77]. In SA, the extent of the implementation of the ABCDEF bundle is not fully understood. A recent nation-wide study found that the prevalence of ICU mobility practice was 47% [78]. However, among the surveyed ICUs, 64% said their staff had never received training to mobilize ICU patients and 55% of these ICUs reported the absence of ICU mobility protocols in SA [78]. The effectiveness and safety of initiating the ICU mobility protocols were only demonstrated in a single disease population (stroke patients) [79]. In addition, severe knowledge gaps and poor attitudes towards pain management were uncovered among ICU nurses, placing pain management at a lower priority [80,81]. The QoL of ICU survivors has mostly been described in developed countries [82,83,84,85,86,87]. A single study reported that QoL in post-ICU Middle East Respiratory Syndrome (MERS) patients was lower than that of MERS survivors who were hospitalized in medical wards [88]. However, ICU rehabilitation, including the ABCDEF bundle or post-ICU rehabilitation service, was not reported in the study.

### 6.2. Future Needs

The mortality rate alone is no longer sufficient to successfully determine the quality of ICU outcomes. Moreover, the magnitude of long-term ICU impairments is increasingly recognized, especially after the COVID-19 pandemic and subsequent global surge in ICU admissions. Hospitals in SA should develop ICU rehabilitation provisions to address the current issues of QoL and functional outcomes in ICU survivors. While a lack of awareness and the consequent proper training for providing critical care rehabilitation are global challenges, Saudi hospitals are no exception. Closing the gap in awareness and knowledge in highlighting the necessity for patients to undergo rehabilitation protocols from the first day of ICU admission is a major challenge to driving ICU culture change. Although there are many other barriers facing ICU rehabilitation, such as staffing, finances, adequate space, and equipment, upholding ICU rehabilitation as a priority would accelerate the momentum towards achieving this mission.

ICUs should prioritize the application of the ABCDEF bundle policy. Effective pain and delirium management, proper use of sedation and analgesia, the application of early mobility and mechanical ventilation, spontaneous weaning trials, and family engagement should be part of ICU daily routine care. The involvement of psychological and social contexts from specialized psychology specialists, social workers, and other ICU practitioners places a greater emphasis on individual patients and their families. This holistic approach to ICU care would provide the foundation for humanizing ICUs in the general attitude of hospitals. Hence, the adaptation of hospital organizations in SA to provide ICU rehabilitation through multi-disciplinary teamwork including ICU physicians, nurses, social workers, clinical psychologists, and psychiatrists and respiratory, physical, and occupational therapists, in addition to proper referrals to other related clinical disciplines, is pivotal for successfully improving ICU survivorship [89].

Facing the aftermath of post-critical illness disability is a major concern for many ICU survivors. Thus, providing post-ICU care is crucial for facilitating the re-emergence of ICU survivors in their communities. Hospitals should identify the return to health status before ICU admission as a plan of care for ICU survivors. Hospitals should adopt referrals of patients to post-ICU clinics and rehabilitation services, taking into consideration the nature of the delayed recovery time of ICU survivors. The provision of rehabilitation services should address the physical, mental, and social challenges that restrain ICU survivors from returning to work and becoming active members of their community.

## 7. Perioperative Rehabilitation

### 7.1. Current Status

Globally, there are millions of major surgeries that are performed with a low risk of postoperative complications [90]. However, some cases exhibit a high risk of developing postoperative complications. This depends on several perioperative factors, such as advanced age, functional dependency, poor exercise capacity, and pre-existing cardiovascular and pulmonary diseases [91]. These risk factors are substantially associated with postoperative complications and an extended length of stay. This negatively impacts long-term QoL, including mobility, self-care, pain, and discomfort. The preoperative evaluation and optimization of these risk factors are essential approaches to improve recovery and minimize postoperative complications, which would improve patient outcomes and QoL in a way that can be measured by recommended instruments such as the ProQOL-5 measure, the EuroQOL-5D quality of life assessment tool, and the WHO Disability Assessment Schedule [76,92,93,94]. Perioperative rehabilitation is required for patients undergoing major surgery, defined as surgery that lasts more than two hours and/or has an anticipated blood loss ≥500 mL [95]. Perioperative rehabilitation encompasses perioperative interventions involving anaesthetists, surgeons, intensivists, nurses, physiotherapists, and respiratory therapists.

Several perioperative approaches, such as optimizing physical activity, pulmonary function, and smoking cessation, are being used to improve recovery and minimize postoperative complications. For example, the effect of preoperative smoking cessation has been well-documented in minimizing postoperative pulmonary complications [96,97]. Of note, it is recommended to start a smoking cessation program at least four to six weeks before surgery [98]. Moreover, preoperative physiotherapy, such as the lung-expansion techniques of breathing exercises, inspiratory muscle training, incentive spirometry (IS), and airway-clearance maneuvers, are well-documented interventions for reducing postoperative complications by enhancing pulmonary function status, thereby potentially improving patient outcomes and QoL [99,100,101,102]. A randomized controlled trial showed that inspiratory muscle training significantly improved inspiratory muscle strength, which was maintained throughout the postoperative period, consequently improving recovery and patient outcomes [103]. Early mobilization is usually prescribed after major surgery to reduce complications and lengths of stay and improve QoL, particularly in daily activities. A systematic review showed that early mobilization can significantly reduce complications and lengths of stay as well as improve functional capacity after cardiac surgery [104].

Currently, standardized perioperative rehabilitation protocols are limited. However, in the United Kingdom, the enhanced recovery after surgery (ERAS) approach integrates different perioperative interventions aimed at improving recovery and minimizing postoperative complications [105]. On the other hand, a simple bundle of respiratory care known as I-COUGH, which includes incentive spirometry, coughing techniques, oral hygiene, getting out of bed, and head elevation, is conducted to minimize the incidence of postoperative complications [106]. The I-COUGH bundle is being incorporated into the ERAS Plus approach and implemented in the United Kingdom [107]. Implementing the I-COUGH bundle was successful in decreasing postoperative pulmonary complications by 50% after major surgery [106].

### 7.2. Future Needs

In SA, the only perioperative rehabilitations being utilized are early mobilization and breathing exercises. This includes providing patients with an incentive spirometry device postoperatively, which is handled by physiotherapists and respiratory therapists. Therefore, a standardized perioperative rehabilitation protocol aimed at improving patient outcomes and QoL is lacking. Implementing a perioperative rehabilitation protocol such as the I-COUGH approach would improve recovery and QoL in SA. A smoking cessation program may be beneficial for improving pulmonary function before surgery. Patient education, specifically about increasing physical activity before surgery, is essential for improving patient outcomes. All these approaches are worth considering for implementation and should be investigated to improve the QoL in patients undergoing surgery.

## 8. Hearing and Speech Rehabilitation

### 8.1. Current Status

Audiology and speech–language pathology (SLP) are related to healthcare professions [108]. Audiology is concerned with the prevention, identification, evaluation, diagnoses, treatment, and management of hearing loss and balance disorders while SLP emphases the prevention, evaluation, diagnosis, treatment, and management of communication and swallowing disorders [109,110]. Patients with hearing and vestibular disorders are managed by audiologists, and patients with speech–language and swallowing disorders are managed by speech–language pathologists through hearing and speech rehabilitation services. Hearing loss and communication disorders have a negative effect on QoL and psychological well-being, causes that are routinely measured by the Satisfaction with Amplification in Daily Living and the Quality of Communication Life Scale evaluations [111,112,113,114]. Approximately 34 million children worldwide have a hearing loss. Congenital, sensorineural hearing loss affects nearly 2–6 newborns per 1000 infants globally, and less-developed countries have a rate of approximately 90% of newborns with hearing loss [115,116]. In SA, the prevalence of different hearing and speech disabilities among Saudi citizens is 3.3% [117]. Mild, moderate, and severe hearing loss represents nearly 1.4% of all citizens in SA, and difficulties with communication and understanding are estimated to affect approximately 10.25% of the total Saudi population with disabilities [118]. Congenital malformations (e.g., hereditary, progressive cochleovestibular anomalies) are responsible for 21.3% of disabilities among Saudi citizens [118,119].

Saudi Vision 2030 fosters improved healthcare services, including hearing and speech rehabilitation, through (a) easing access to these services, (b) improving the value of these services, and (c) reinforcing the prevention of health threats [120]. Therefore, initiatives were carried out by the government to prevent such disabilities and reduce their effects. For instance, the national newborn hearing screening (NHS) and the early intervention of hearing impairment programs were implemented by the Saudi government [121]. The first phase of the NHS was initiated by the Ministry of Health, covering more than 60% of newborns in 30 referral hospitals [122]. Subsequently, the NHS service has expanded to reach approximately 89% of newborns [123].

Diagnosing hearing loss and fitting hearing aids (HAs) are the foremost services conducted by audiologists in SA. To enable those with a hearing loss to use auditory signals effectively and perform better speech communication, the fitting of hearing aids (HAs) is necessary for eligible patients [124]. However, the majority of audiologists (98.2%) either rarely or never use real ear measurements and mainly depend on functional gain and the patients’ feedback to confirm that fit of HAs [125]. Fitting HAs by using verification measures with validated prescriptive targets was related to improving patient outcomes [126]. The shortage of audiology services also includes tinnitus and vestibular assessment and rehabilitation, auditory processing disorder assessments and management, aural re/habilitation services, and services geared towards hearing conservation and protection [127].

SLPs provide services relating to verbal communication skills, including diagnosing and managing communication disorders, improving speech, and managing stuttering. In addition to improving receptive language skills, these areas of practice are the main daily responsibilities of practicing SLPs in SA. According to Alanazi, only 27% of SLPs worked with dysphagic patients in SA [127]. It is estimated that approximately 32.4% of practicing SLPs in SA work with patients with voice and resonance disorders [127]. Whether the referral is required or not depends on where the audiologists and SLPs practice. Since most audiologists and SLPs in SA work in a healthcare setting, referrals from physicians are mandatory [128]. However, patients would be able to see either professional without a referral from a physician or other healthcare provider through a “direct access” system [19]. For details about audiology and SLP services in SA, readers are referred to reference [127].

The dearth of audiology and SLP services in many Saudi regions, except in major cities (e.g., Riyadh and Jeddah), was reported in the literature [127,129]. The situation of most undergraduate and graduate academic programs in Riyadh and Jeddah could be the reason [127,129]. Additionally, the following explanations might elucidate the lack of some specialized audiology and SLP services in SA. First, most practicing audiologists and SLPs have bachelor’s degrees with a few years of clinical experience. Second, there is a scarcity of standardized Arabic tests (e.g., APD tests) and specialized training courses in these areas. Third, several healthcare settings are not equipped with the necessary equipment (e.g., electrophysiological and vestibular equipment). Finally, some of these settings do not have specially designed programs (e.g., programs for hearing protection, aural re/habilitation, or other specialized services) [127].

### 8.2. Future Needs

Many audiologists and SLPs reported the need for a better understanding of both fields among other healthcare professionals and in the community, as well as a better recognition of the professions by the national authorities [127]. Although it was found that there exists a reasonable level of public awareness of audiology and SLP services in SA, there is still a need for the public to be informed about these available services [128]. Guidelines, policies, continuing education, national databases, and statistics represent other audiology and SLP demands [127]. Audiology and SLP services should be extended to reach rural areas in all Saudi regions. The use of telepractice at any time, and particularly during times such as the COVID-19 pandemic, is one of the solutions to overcome the shortage of such services in these areas [130]. It was found that 93% of SLPs in SA had started offering telehealth services during COVID-19 [131]. Collaborative efforts between audiologists and SLPs, the Saudi Society of Speech-Language Pathology and Audiology, audiology and SLP academic programs, and other stakeholders (e.g., the Ministry of Health and Ministry of Education) to continuously work towards improving hearing and speech rehabilitation are crucial.

## 9. Substance Use Disorder Rehabilitation

### 9.1. Current Status

Substance use disorder (SUD) is a global health issue, and SA is no exception [132]. SUD is a chronic, relapsing condition characterized by maladaptive patterns of substance use associated with impartment or distress that affects physical and psychological well-being [133]. Several forms of substance use have been reported in SA, including tobacco, hashish, alcohol, and narcotics [134]. For example, the Saudi national surveillance of tobacco use in 2019 (Global Adult Tobacco Survey) revealed that 19.8% overall (30.0% of men and 4.2% of women) currently use tobacco [135]. Among inpatients being treated for SUD in 2017, hashish was the most commonly used substance (56.4%), followed by amphetamines (53.1%), alcohol (29.3%), and opioids (19.4%) [134]. However, the prevalence of SUD varies based on sex, due to social and cultural structure, and based on geographical region, due to the border territories of trafficking roads [134,136,137]. SUD was found to be greater among Saudi men and in the eastern, southern, and northern regions of the nation [134,136].

SUD is highly associated with mental disorders and social stressors [138]. In SA, three studies reported that the use of amphetamines and alcohol is associated with suicidal ideation, depression, psychosis, neuroticism, and anxiety [139,140,141]. Furthermore, unemployment, peer pressure, and family history were associated with relapse and failure to remain abstinent from substance use following detoxification and discharge from a rehabilitation program [136,142]. SUD treatment seeking is not precisely intended to quit such substance use, but ultimately to avoid its negative consequences and improve wellbeing and QoL [133]. The complexity of SUD makes the consideration of QoL greatly substantial to integrate a full range of problems associated with SUD, such as physical, mental, social, vocational, and residential status, which can be measured by different instruments such as the multidimensional index of life quality, ProQOL-5 measure, and the 12-Item Short Form Survey [92,133,143,144].

Globally, studies have revealed that individuals with SUD self-reported poorer QoL than the general population [145,146,147]. Indeed, QoL was a predictor of substance use characteristics, treatment readiness, and abstinence outcomes [148,149,150]. For example, tobacco use was found to be negatively associated with QoL, which predisposed smokers to greater impairment, disability, and relapse from treatment [151]. The magnitude of QoL impairment was even greater among users of narcotics [149]. When compared to the general population, those who were substance-dependent had a significantly poorer QoL across several domains, including physical and mental functioning [152,153,154]. In SA, there is scarce literature about the intersection of QoL and SUD; however, three studies reported the associations between SUD and psychological disorders that are known to affect QoL [139,140,141]. Notably, state officials have allocated considerably comprehensive prevention and treatment services, provided by physicians, psychologists, and social-service specialists, to those affected by SUD [155,156].

SA’s efforts to tackle SUD are allocated to prevention and treatment services that are both regulated and operationalized under the umbrella of the National Committee for Tobacco Control and the National Committee for Narcotics Control, which were founded by the Council of Ministers and are composed of representatives of several ministries [157,158]. SUD prevention services are exemplified by the adoption of international policies and regulations supported by the World Health Organization and the United Nations Commission on Narcotic Drugs to prevent the uptake of substance use among youth and to promote awareness through mass media and public outreach campaigns [159,160]. In addition, SA provides free SUD treatment from the Ministry of Health in different modalities. First, treatment from tobacco use is supported by walk-in, mobile, and telehealth clinics that primarily provide pharmaceutical therapy from tobacco use [161]. However, behavioral therapy during and after treatment seems suboptimal to prevent relapse and improve QoL among tobacco users [162]. Second, treatments for hashish, alcohol, narcotics, and other substances are provided by a comprehensive therapy program in inpatient health facilities (e.g., the Eradah (Al Amal) Mental Health Complex) and a therapeutic community for long-term therapy, which is a drug-free residential setting that integrates several elements of rehabilitation such as physical, mental, social, and spiritual functioning [134,163]. Furthermore, the prevention and treatment of SUD are partially provided by several non-profit organizations across the nation which are licensed by the Ministry of Labor and Social Development [134,163].

### 9.2. Future Needs

SUD prevention and treatment are considered comprehensive in SA. However, the integration of QoL elements into rehabilitation is not optimal. There are still areas for improvement in optimizing treatment, preventing relapse, and improving QoL. The first is the adoption of an integrated, multisystem, recovery-oriented model in the treatment of SUD to address its complexity and chronicity [133]. For example, elements such as family partnership, community engagement, employment opportunities, and other elements are pivotal in increasing the odds of successful treatment and recovery [133]. Second, long-term follow-up and monitoring are warranted to prevent relapse and improve abstinence [164]. This seems to be less considered in tobacco use than in other substance use, even though relapse and the concurrent use of other substances with tobacco are highly prevalent [165,166]. Third, assessing QoL domains in the rehabilitation program before, during, and after treatment to evaluate the patient’s vulnerability and treatment goals is key, and has been proven to predict substance use initiation and abstinence outcomes [148,149,150]. Future studies are needed to optimize SUD treatment and meet the nation’s vision for improving health and QoL.

## 10. Vocational Rehabilitation

### 10.1. Current Status

Rehabilitation services support individuals with disabilities to be independent and participate actively in their society, which in turn helps them become employed and enhances their QoL [167]. Vocational rehabilitation (VR) provides individuals with all types of disabilities with better opportunities for employment [168]. In general, the aim of VR is to help those with disabilities to find or return to employment after a disabling injury [168]. The benefits of undergoing a VR program include improving income and job satisfaction, increasing employment opportunities, enhancing employment mobility, and optimizing skills [169]. Different studies have revealed that VR qualifications significantly expand the opportunities of individuals with disabilities to not only retain their employment positions but also to obtain employment in other areas [170,171]. After completing a VR program, the probability of employment increases as a result of overcoming perceived obstacles associated with a disability [172]. Several instruments have been recommended to measure patients’ QoL in VR programs. These include, but are not limited to, the WHO Disability Assessment Schedule and The International Classification of Functioning checklist [173].

In general, more than half of those with disabilities who completed a VR program find employment [172]. Employment was strongly associated with QoL in individuals with disabilities [174]. Thomas, Bax, and Smyth (1991) reported that individuals with disabilities who are socially isolated encountered difficulties with social skills. Social engagement for individuals with disabilities is pivotal to increasing their independence and social participation [175]. Therefore, employment can help those with disabilities to integrate into society, which in turn enhances their QoL.

The Ministry of Human Resources and Social Development in SA is the entity that regulates and the supports VR programs in the country [176]. Such programs are designed to rehabilitate individuals with physical disabilities to adapt to their new conditions. In addition to the government entity, the private sector contributes substantially in providing VR programs to individuals with disabilities [176]. Moreover, VR accommodates students with disabilities in the educational sector by providing them with full support of educational, social, and psychological services to ensure that individuals with disabilities have a good QoL [176].

### 10.2. Future Needs

The development of comprehensive policies related to health rehabilitation programs should integrate VR as a fundamental component to help individuals with disabilities become independent. As Saudi universities currently do not offer educational certificates in VR, other related degrees, such as occupational therapy, should integrate VR components into their curricula. This would provide occupational therapists with a wider scope for dealing with individuals with disabilities. Moreover, the Ministry of Education should form an initiative to utilize recent research in the field of VR to develop VR-related curricula for inclusion in educational rehabilitation programs in Saudi universities. The Ministry of Human Resources and Social Development should also create policies and initiatives that enhance employment preparation programs to encourage individuals with disabilities to become employed and independent, thereby providing opportunities for them to be active community members. 

VR in SA is considered suboptimal, and future work should seek improvements in VR services from program components, the allocation of services in healthcare facilities, and the integration of VR into the educational sector. Factors related to increasing awareness and policy development from different government sectors to empower VR program referrals and outcomes are highly warranted. In addition, the long-term follow-up of individuals with disabilities will assess their societal and occupational status in the future.

## 11. Conclusions

Health rehabilitation services are fundamental components of SA’s vision of promoting a high QoL, especially among individuals with chronic diseases. This review is one of the first to address the current status and future needs of comprehensive health rehabilitation services provided in SA. Experts in different sectors of health rehabilitation across the country have illustrated that the current health rehabilitation services are suboptimal for the individuals involved in health rehabilitation programs. To optimize the QoL of individuals with chronic diseases, the future demands of health rehabilitation services should be tailored based on the unique requirements of each rehabilitation service and the patients it serves (Figure 1).

Overall, the shared need to improve all health rehabilitation services in the future include, but are not limited to, the adoption of home-based and telehealth services, the integration of multi-governmental sectors, the empowerment and allocation of health rehabilitation specialists, public awareness campaigns, policy legislation and guideline development, and the implementation of a long-term follow-up system (Figure 1). Urgent and holistic actions are paramount to address the pressing need to optimize health rehabilitation services in SA. To implement the recommendations of this study, the effective communication of these needs should be adopted by national health policymakers to enact necessary laws and regulations. The design of programs and guidelines for implementation should be communicated to institutional and managerial levels for action, monitoring, and evaluation.

Although this study is based on the health rehabilitation services in SA, the experts’ recommendations may be applicable to the wider context of other countries’ health systems as health rehabilitation services are not well optimized globally. The assessments and recommendations presented in this study are limited based on the experts’ opinions, which are subjective to the selection bias of the literature and lack explicit criteria for literature selection and appraisal to mitigate bias. Future researchers shall seek to objectively assess health rehabilitation services in SA from different stakeholders, including policymakers, clinicians, and patients.

## Figures and Tables

**Figure 1 healthcare-11-00389-f001:**
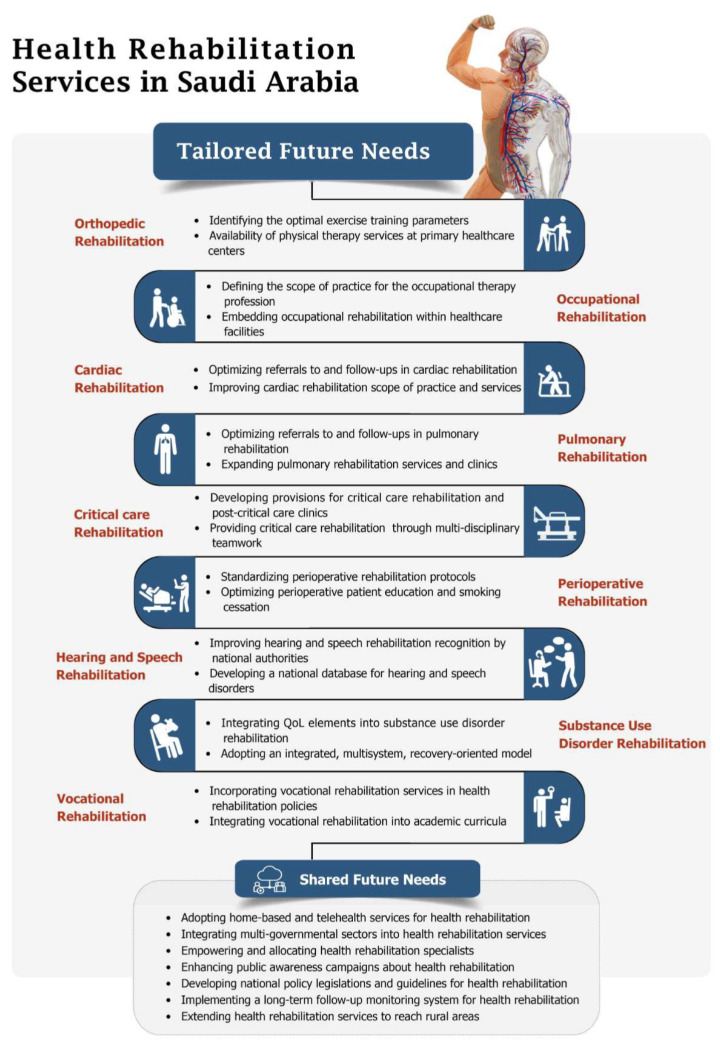
Tailored and shared future needs of health rehabilitation services in Saudi Arabia.

## Data Availability

Not applicable.

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
