# Peer review of "The Intersection of Health Rehabilitation Services with Quality of Life in Saudi Arabia: Current Status and Future Needs"

_healthcare, 2023, doi:10.3390/healthcare11030389_

Round 1
Reviewer 1 Report
The title is longer, it is difficult to understand the goal of the review.
Which kind of health professionals are involved in each health rehabilitation services?
Which kind of patients and illness are involved in each health rehabilitation services?
Line 122-143: This general information could include in background section. I will include information/results only about Orthopedic rehabilitation
How evaluate the quality of life in each health rehabilitation services?
In the future needs, the authors could compare between the SA health rehabilitation services versus the health rehabilitation services of other countries.
Pulmonary rehabilitation only focuses on COPD, however other respiratory diseases need pulmonary rehabilitation, why didn't it include?
Author Response
Response: Dear reviewer, on behalf of my co-authors, we would like to thank you for your valuable feedback and input into our manuscript. We believe that your feedback has an insightful contribution to this work. Please, find below point by point reply and edits accordingly.
- The title is longer, it is difficult to understand the goal of the review.
Response: The title has been changed to:
“The Intersection of Health Rehabilitation Services with Quality of Life in Saudi Arabia: Current Status and Future Needs”
2.a. Which kind of health professionals are involved in each health rehabilitation services?
2.b. Which kind of patients and illness are involved in each health rehabilitation services?
Response: For both comments, we have added/highlighted the health professionals are involved and the kind of patients and illnesses involved in each health rehabilitation service of the manuscript as follows:
Orthopedic rehabilitation:
Line: 113-115
“Orthopedic rehabilitation services can include a multidisciplinary team of physical and occupational therapists to provide efficient and relatively specialized rehabilitation.”
Line: 121
“In SA, musculoskeletal conditions are escalating rapidly, ranking in the top three potential reasons to visit the hospital and accounting for 38% of family practice visits”
Occupational rehabilitation:
Line:164
“Occupational therapy is a professional health practice that is an essential part of the health rehabilitation team. Occupational therapists apply the therapeutic use of everyday life occupations to persons, groups, or populations to enhance occupational performance and participation”
Line: 170-171
“Habilitation, rehabilitation, and promotion of physical and mental health wellness for individuals with all levels of occupational issues, such as chronic pain, sport and work injuries, and congenital and developmental disorders, are the main services that occupational therapy provides”
Cardiac rehabilitation:
Line: 207-2010
“In SA, the estimated age of patients with acute and chronic heart failure was 10 years younger than patients in other developed countries coupled with high-risk factors of cardiometabolic diseases such as coronary artery disease and diabetes mellitus”
Line: 223-225
“In SA, the number of cardiac rehabilitation programs is low, and they are mostly run by cardiologists and physiotherapists to manage patients with heart failures and heart transplants.”
Pulmonary rehabilitation:
Line: 241-244
“Pulmonary rehabilitation is an interdisciplinary program that is mostly run by pulmonologists, physiotherapists, and respiratory therapists for patients with chronic pulmonary diseases such as chronic obstructive pulmonary disease (COPD), interstitial lung diseases, bronchiectasis, asthma, and scoliosis.”
Critical care rehabilitation:
Line: 304-305
“Acute respiratory distress syndrome, sepsis, and acute kidney injury are examples of critical illnesses that are treated in the ICU”
Line: 364-366
“the adaptation of hospital organizations in SA to provide ICU rehabilitation through multi-disciplinary teamwork such as ICU physicians, nurses, respiratory, physical, and occupational therapists, social workers, clinical psychologists, psychiatrists and proper referral to other related clinical disciplines is pivotal for successfully improving ICU survivorship”
Perioperative rehabilitation:
Line: 389-393
“Perioperative rehabilitation is required for patients undergoing major surgery which is defined as surgery that lasts more than two hours and/or with anticipated blood loss ≥ 500 ml [96]. Perioperative rehabilitation encompasses perioperative interventions, involving anaesthetists, surgeons, intensivists, nurses, physiotherapists, and respiratory therapists.”
Hearing and Speech Rehabilitation:
Line: 439-442
“Patients with hearing and vestibular disorders are managed by audiologists and patients with speech-language and swallowing disorders are managed by speech-language pathologists in hearing and speech rehabilitation services.”
Substance Use Disorder Rehabilitation:
Line: 546-548
“Of note, state officials, have allocated considerably comprehensive prevention and treatment services provided by physicians, psychologists, and social service specialists for those affected by SUD”
Vocational rehabilitation:
Line: 587-588
“Vocational rehabilitation (VR) provides individuals with all types of disabilities with better opportunities to become employed”
Line: 618-621
“As Saudi universities currently do not offer degrees in VR, other related degrees, such as occupational therapy, should integrate the VR components into their curricula. This would provide occupational therapists with a wider scope for dealing with individuals with disabilities”
- Line 122-143: This general information could include in background section. I will include information/results only about Orthopedic rehabilitation
Response: It has been modified to be included in the background section instead of the orthopedic rehabilitation section in the following lines:
84-87: “QoL is influenced by potential factors, which could be extrinsic, such as environmental and social factors, or intrinsic, such as individual factors (e.g., age, sex, body weight, mental health), lifestyle behavior (e.g., functional independence), health condition, and education [14].”
- How evaluate the quality of life in each health rehabilitation services?
Response: We have added/highlighted the quality-of-life measures in each health rehabilitation service as follows:
Orthopedic rehabilitation:
Line: 138-140:
“Patients reported outcomes including the generic tools related to specific aspects of individuals' health are suggested to be used as a long-term follow-up for enhancing the recovery of the patients in rehabilitation”
Occupational rehabilitation:
Line: 180-183
“Occupational rehabilitation services have demonstrated improvements in QoL in various healthcare settings which are routinely measured by different instruments such as Barthel Index, Functional Independence Measure, The Canadian Occupational Performance Measure, and The Lawton IADLs scale”
Cardiac rehabilitation:
Line: 217-220
“There are recommended instruments to monitor patients’ improvements during and after cardiac rehabilitation programs which have been reflected in patients’ QoL such as the Borg Rating of Perceived Exertion Scale, multiple-repetition maximum test, 6–minute walk test, and incremental shuttle walk test”
Pulmonary rehabilitation:
Line: 251-256
“In addition, pulmonary rehabilitation programs decrease dyspnea, increase exercise capacity, improve HR-QoL, and reduce the risk of hospitalizations through exercise training, psychosocial and nutritional interventions, and education and smoking cessation [52]. Patients’ outcomes that reflect QoL in pulmonary rehabilitation programs are routinely measured by a 6-minute walk test, Chronic Respiratory Disease Questionnaire, St George’s Respiratory Questionnaire, and 36-Item Short Form Survey”
Critical care rehabilitation:
Line: 326-328
“Of note, there are different instruments that are adapted to measure QoL among ICU survivors such as the 36-Item Short Form Survey and the EuroQOL-5D quality of life assessment tool”
Perioperative rehabilitation:
Line: 385-389
“Preoperative evaluation and optimization of these risk factors are essential approaches to improve recovery and minimize postoperative complications, which would improve patient outcomes and QoL that can be measured by recommended instruments such as the ProQOL-5 measure, the EuroQOL-5D quality of life assessment tool, and WHO Disability Assessment Schedule”
Hearing and Speech Rehabilitation:
Line: 443-444
“Hearing loss and communication disorders have a negative effect on the QoL and psychological well-being causing that are routinely measured by Satisfaction with Amplification in Daily Living and the Quality of Communication Life Scale”
Substance use disorder rehabilitation:
Line: 533-536
“The complexity of SUD makes consideration of QoL greatly substantial to integrate a full range of problems associated with SUD such as physical, mental, social, vocational, and residential status that can be measured by different instruments such as the multidimensional index of life quality, ProQOL-5 measure, and 12-Item Short Form Survey”
Vocational rehabilitation:
Line: 597-599
“Several instruments have been recommended to measure patients’ QoL in VR programs that include but are not limited to the WHO Disability Assessment Schedule, and The International Classification of Functioning checklist”
- In the future needs, the authors could compare between the SA health rehabilitation services versus the health rehabilitation services of other countries.
Response: We are in line with you with respect to the importance of comparing future needs in SA with health rehabilitation services in other countries. In addition, we believe that such future needs might be shared with the global needs for health rehabilitation services as stated in the background. However, the scope of this study is stating the future needs specific and tailored to SA since the panel of experts here are practicing and familiar with the unique needs pertaining to the SA healthcare system which makes future needs in other countries outside of the scope of this manuscript.
- Pulmonary rehabilitation only focuses on COPD, however other respiratory diseases need pulmonary rehabilitation, why didn't it include?
Response: We have added statements related to other chronic pulmonary diseases in SA as follows:
Line: 271-274
“On the other hand, limited studies investigated the uptake of pulmonary rehabilitation in other chronic pulmonary diseases in SA. AlMoamary M.S. (2012), however, reported promising outcomes for a limited number of patients with interstitial lung diseases, bronchiectasis, asthma, and scoliosis after enrollment in a pulmonary rehabilitation program in SA”
Reviewer 2 Report
Dear Authors,
thank you for the opportunity to read your manuscript.
The results are interesting and usefull to underline specific national policies concerning rehabilitation.
I would like to suggest to modify the text in order to guarantee a structure of the Current Status, able to present the general epidemiological context related to the diseases of interest, than introduce the specific SA context, finally explaining the QoL measures or factors impacting on the patients and being of interest for the national SA program.
I kindly suggest to look at the additional file provided in order to review the text point by point.
Good luck!

Author Response
Dear Authors,
thank you for the opportunity to read your manuscript.
The results are interesting and useful to underline specific national policies concerning rehabilitation.
Response: Dear reviewer, on behalf of my co-authors, we would like to thank you for your valuable feedback and input into our manuscript. We believe that your feedback has an insightful contribution to this work. Please, find below point by point reply and edits accordingly.
- I would like to suggest to modify the text in order to guarantee a structure of the Current Status, able to present the general epidemiological context related to the diseases of interest, than introduce the specific SA context, finally explaining the QoL measures or factors impacting on the patients and being of interest for the national SA program.
Response: Since we have addressed comprehensive health rehabilitation services in SA and each health rehabilitation is uniquely different in its coverage of multiple diseases and its serving patients, following the same order will not completely feasible for all health rehabilitation services. However, my co-authors and I believe each health rehabilitation service covers the aspects of the general epidemiological context globally and in SA as literature permits in addition to its relationship with QoL. This will be highlighted in the manuscript for each section.
- Abstract page 1 line 38 Modify chronic disease in plural
Response: it has been modified to “chronic diseases”
- Line 40 The problem defined is not only related to SA but also concerned the countries all over the world. Please add this point!
Response: It has been added to the following sentence
Line: 41-43
“Health rehabilitation services in SA like in other countries are suboptimal for individuals in health rehabilitation programs
- Line 45 the unique requirement of each service or the unique requirement of each need? It depends also to the need of the populations!
Response: it has been modified as follows:
“To optimize the QoL of individuals with chronic diseases, health rehabilitation services should be tailored based on the unique requirements of each service and its serving patients”
Section Background
- Line 57 chronic illnesses! In my opinion modify in plural
Response: it has been modified to “chronic illnesses”
- Line 61 value-based care, please, modify in value-based healthcare! Please verify all the manuscript and modify accordingly!
Response: it has been changed to “value-based healthcare”
- Line 62 and 63 The value-based healthcare principals required to introduce only preventive and healthcare program in general able to guarantee the financial sustainability, and effectiveness, but with a correct balance with the patients reported outcome measures. Please add some considerations related to the correct definition of this principle, referring to the Michael
Porter manuscripts
Response: it has been modified as follows
Line: 60-62
“This program is based on the principle of value-based healthcare, which ensures transparency and financial sustainability in line with improving patient outcomes by applying a new model of care related to disease prevention.”
- Line 74 The ADL measure is one of the factors influecing the quality of life. The scales measuring QoL always consider this factor, but the way in which the ADL was measure could be different. Please, rephrase the sentence in order to better explain this concept to be more in line with what you have declared in line 77-81, that is more appropriate!
Response: it has been modified as follows:
Line: 69-70
“Most long-term conditions, especially those with multimorbidity, have no complete cure or recovery of physical, psychological, or social well-being [7], which may potentially compromise QoL. Therefore, long-term rehabilitation is the core of patient care to enhance well-being and QoL.”
- Line 89-90 the range 10-30% of the total hospital admissions seems to be very large! If it is possible, add also the consideration related to the fact that not only hospital admissions but also the specific healthcare problem could impact on the need of rehabilitation activities. This percentage is consistent with the impact on other countries or healthcare system? If the answer is no, why? Please add some explanation to define if this result could be considered replicable or not! This could be consistent also with the objective of the study: in line 107-108 the authors declared that the results could be replicable to the global context. In this view it is necessary to define how the context described could be generalizable.
Response: After critically reviewing this statement, we believe that it would be better to omit such a statement to be careful in describing the health rehabilitation services in SA. This sentence has been deleted and the logic of writing has not been compromised. However, the emphasis on the need for optimizing health rehabilitation services in SA and globally still presents in the context of the background.
Line: 86-94
“Rehabilitation services that integrate diet, therapeutic exercise, behavioral modification, and other components have been shown to optimize health functioning, improve symptoms, enhance HR-QoL, and reduce healthcare utilization [15, 16]. Recently, the Health Sector Transformation Program was established as part of the Kingdom's Vision 2030 strategy. The program's goals include expanding access to health care services by providing universal coverage, promoting geographic equity, and expanding the use of electronic medical records and other forms of e-health.”
- Orthopedic rehabilitation Section It is not clear and not defined which are the factors able to measure specifically the QoL in orthopedic rehabilitation. Please add this information that is absolutely necessary in order to link the results with the future needs section!
Response: The orthopedic rehabilitation section has been modified as follows:
Line: 131-143
“Four-dimensional factors include MSK pain, physical function, mental health, and self-perception of health have an association with the QoL [18]. The quality of the rehabilitation services provided is a key measure of the patient's QoL, and currently, there is a growing interest in measuring HR-QoL. For instance, osteoarthritis is the most prevalent condition in SA and is one of the primary causes of pain and disability worldwide[21]. Physical therapy is the first line of intervention, and physical therapy interventions, including aquatic physical therapy [14, 22], or overground training [23], showed a substantial improvement in patient QoL. Patients reported outcomes including the generic tools related to specific aspects of individuals' health are suggested to be used as a long-term follow-up for enhancing the recovery of the patients in rehabilitation. Therefore, the provision of QoL measures in physical therapy outcome research has been augmented. The overarching goal of physical therapy is to ensure pain-free movement, improve function, and to implement psychosocial support to improve patients' health and overall well-being.”
Line: 145-150
“Assessing QoL in orthopedic physical therapy is essential for evaluating patient well-being and intervention efficacy. The major limitation of the QoL measure is the lack of sensitivity; therefore, a patient centered QoL measurement can be established to address all the factors that impact recovery and perceptions of care. Furthermore, despite the widespread provision of orthopedic physical therapy for movement-related impairment, few studies have focused on the optimal parameters for exercise interventions designed to improve QoL.”
- Future needs Line 189 the number of occupational therapists is relatively low. This sentence needs to be supported by numbers or evidence. Which is the ration among population and occupational therapists all over the world? Or in some countries similar to SA? To add this information could be essential in order to quantify the problem, and also to generalize the results to other international context.
Response: It has been modified as follows:
Line: 194-198
“the number of occupational therapists was relatively low. This issue is consistent with other countries such as the United States and Norway as their demands for occupational therapists are increasing [35-37]. Indeed, it was estimated that occupational therapy had the national greatest need among healthcare professionals in SA, and 137 full-time equivalents are needed to meet the national manpower of occupational therapists in the nation”
- Current Status Line 197-199 Sometimes the authors give an overview of the clinical problem in SA and in other sections is not clear the impact of the specific healthcare problem in SA. In this case a description concerning the cardiac rehabilitation and the specific SA context is provided. The impact of cardiovascula diseases in SA is missing. In this section it also not clear which measures or which factors could be considered in order to empower the QoL of the patients and also to improve the effectiveness of the rehabilitative intervention.
Response: it has been modified as follows:
Line: 207-210
“In SA, the estimated age of patients with acute and chronic heart failure was 10 years younger than patients in other developed countries coupled with high-risk factors of cardiometabolic diseases such as coronary artery disease and diabetes mellitus [41].”
Line: 217-220
“There are recommended instruments to monitor patients’ improvements during and after cardiac rehabilitation programs which have been reflected in patients’ QoL such as the Borg Rating of Perceived Exertion Scale, multiple-repetition maximum test, 6–minute walk test, and incremental shuttle walk test [46]”
- Current Status Line 241-243 In this case is very well described the SA context, but it is not clear if these results are in line with the international epidemiological data or are different, specifying also if the considerations are generalizable or specific for the SA national context.
Response: it has been modified as follows:
Line: 261-263
“Although the prevalence of COPD in SA is lower than in other developed countries, the prevalence of COPD from 1990 to 2019 in SA increased by 49%, from 1,381.26 cases per 100,000 people to 2,053.04 cases per 100,000 people”
- Also in this section, it is not clear which measures or which factors could be considered to empower the QoL of the patients and also to improve the effectiveness of the rehabilitative intervention. For example the 6 minutes walking test is normally used for the patients affected by polmunary diseases
Response: it has been modified as follows:
Line: 251-256
“In addition, pulmonary rehabilitation programs decrease dyspnea, increase exercise capacity, improve HR-QoL, and reduce the risk of hospitalizations through exercise training, psychosocial and nutritional interventions, and education and smoking cessation [52]. Patients’ outcomes that reflect QoL in pulmonary rehabilitation programs are routinely measured by a 6-minute walk test, Chronic Respiratory Disease Questionnaire, St George’s Respiratory Questionnaire, and 36-Item Short Form Survey [53-55]”
- Current Status Line 274-276 In this section the epidemiological context is not described and it is missing. If the authors would consider any kind of healthcare problem leading to an ICU recovery, or specific diseases would be considered.
Response: It has been modified as follows:
Line: 302-305
“Generally, ICU critical illnesses are clinical syndromes and heterogeneous by nature. Thus, making the reliability of critical illness epidemiological data unavailable and challengingly to obtain. Acute respiratory distress syndrome, sepsis, and acute kidney injury are examples of critical illnesses that are treated in the ICU [65, 66]..”
- Current Status Line 398-400 In this section the epidemiological context is described for th SA national context, but it is not clear the prevalence and incidence in other international countries. Please add this relation, describing the differences or the links
Response: it has been added as follows:
Line: 445-448
“Globally, approximately 34 million children around the world have hearing loss [116]. Congenital sensorineural hearing loss affects approximately 2–6 per 1000 newborns in low-, middle-, and high-income countries[117, 118]. At least 90% of newborns with hearing loss live in less developed countries[118].”
- Current Status Line 486-488 The measure of QoL is reported with international references, but it is not clear if the epidemiological SA context is generalizable or not. Please add some considerations in order to better define the context.
Response: it has been modified as follows:
Line: 544-546
“In SA, there is scarce literature about QoL intersection with SUD; however, three studies reported the associations between SUD and psychological disorders that are known to be affecting QoL [141-143]”
- Conclusion must be improved, in particular defining the future steps in terms of actions and prioritization of these actions. The SA Ministry of Health would start with one specific area of interest? Please specify and declare the limitations. I guess that some managerial implications could be added: for example in different settings a lack of professionals is declared. What we could do to solve this problem?
Response: It has been modified as follows:
Line: 649-653
“Urgent and holistic actions are paramount to address the pressing need to optimize health rehabilitation services in SA. To implement the recommended future plans of this study, effective communication of these needs should be adopted by health national policymakers to enact necessary laws and regulations. Designing programs and guidelines for implementation should be communicated to institutional and managerial levels for action, monitoring, and evaluation.”
Line: 657-662
“The assessments and recommendations presented in this study are limited based on the expert’s opinion which is subjective to the selection bias of literature and lack explicit criteria for literature selection and appraisal to mitigate bias. Future researchers shall seek to objectively assess health rehabilitation services in SA from different stakeholders including policymakers, clinicians, and patients.”
Good luck!
Response: Thank you. It is much appreciated.
Reviewer 3 Report
Thank you for the opportunity to review this article, which is a review of the current status of health rehabilitation service in Saudi Arabia. The article is very comprehensive, covering and discussing multiple areas of health rehabilitation. The contents are very narrative, yet analyzed and presented rigorously. The editorial requirements are met and English language is fine. No major spell typos detected. Overall, the review provides with a comprehensive picture of the state of the art of health rehabilitation services and, for each area, describes the future perspectives of development. This article can be a useful informational tool for countries and entities wishing to undertake similar reviews of current practices with a view to future implementations.
Author Response
Thank you for your feedback. It is much appreciated.
Round 2
Reviewer 2 Report
Dear Authors,
Thank you very much for your effort in modifying the text adding all the comments and suggestions proposed.
I have no other comments to be adressed.
Good luck